# The impact of Daylight Saving Time on dog activity

**Lavania Nagendran**[1]☉*, **Ming Fei Li**[2]☉, **David R. Samson**[1], **Lauren Schroeder**[1]

**1** Department of Anthropology, University of Toronto Mississauga, Mississauga, Ontario, Canada,
**2** Department of Anthropology, University of Toronto, Toronto, Ontario, Canada

☉ These authors contributed equally to this work.
* lavania.nagendran@mail.utoronto.ca

**Data Availability Statement:** All relevant data are within the manuscript and its Supporting Information files.

**Funding:** This research was supported through Discovery Grants from the Natural Sciences and Engineering Research Council of Canada (RGPIN-

## Abstract

While most studies on Daylight Saving Time (DST) focus on human sleep and well-being, there is a dearth of understanding of how this sudden, human-mitigated change affects the routines of companion animals. The objective of this study was to assess how DST influenced the morning activity pattern of dogs (*Canis familiaris*). We used accelerometers to record activity in 25 sled dogs and 29 caregiver-companion dog dyads located in or near Ontario, Canada during the Fall Back time shift. We looked at morning onset activity based on sunrise time (both groups), handler arrival time (sled dogs), and caregiver Got-up Time (companion dogs), and compared pre-DST measures with the three days following DST. We found that sled dogs were less active around sunrise in all post-DST days compared to pre-DST, as sunrise coincided with handler arrival time before DST but not after. Companion dogs showed no change in morning activity based on sunrise times before and after DST. On the Sunday of DST, sled dogs were more active before handlers arrived, but their activity returned to pre-DST levels in the following two days. Caregiver and companion dog activities did not change on the day of DST. After DST, caregivers woke up earlier on weekdays, but companion dogs maintained their pre-DST activity patterns. Overall, we found that sled dogs took one day to adjust to a change in handler arrival time yet neither companion dogs nor their caregivers showed any morning activity difference after the Fall Back DST transition. In summary, our findings highlight the importance of flexible routines and gradual changes in helping dogs adjust to abrupt schedule modifications, offering valuable insights for optimizing dog care practices during time shifts.

## Introduction

Daylight Saving Time (DST) occurs twice a year in some countries and is characterized by an hour change in local clock time primarily to align daylight hours with human activity periods [1]. In the Northern Hemisphere, the clock time is pushed forward in the spring between March and April (Spring Forward) and pushed back in the fall/autumn between September and November (Fall Back). Harrison's [1] review of scientific literature examining the influence of DST on human sleep and behaviour suggests that disruptions in sleep continuity (e.g.,

2020-04159 to L.S. and RGPIN-2020-05942 to D. R.S).

**Competing interests:** The authors have declared that no competing interests exist.

increased sleep fragmentation and intra-daily variability) and efficiency may last up to a week after the shift. Individuals with extreme chronotypes, which would be an individual's inclination to sleep much later or rise much earlier (extreme "evening" and morning" types, respectively), experienced greater difficulty adjusting to the time change. This was also observed among individuals who reported poor sleep quality prior to the time shift. In a longitudinal study spanning more than two decades, Varughese and Allen [2] found that increased fatal traffic accidents were reported following both DST shifts. The authors suggested that physiological adjustments relating to sleep deprivation resulted in higher fatalities in the spring, while the fatalities following the Fall Back transition were associated with behavioural changes anticipating the additional hour of sleep (e.g., staying out later the night before, driving drowsy or under the influence of alcohol). In summary, there is considerable evidence that DST has a substantial impact on human sleep and behaviour.

Recognizing that DST is an extreme form of anthropogenic intervention on the effects of natural light on circadian rhythm regulation [3, 4], we aim to investigate how this abrupt shift in the timing of human activity affects companion animals. This paper will be the first to examine how DST influences the activity of dogs (*Canis familiaris*). Although research on this topic is limited, inferences can be made based on what is known about dog activity patterns. Dogs are highly entrained to human behaviour. For instance, Griss and colleagues [5] found that the activity pattern of companion dogs was more correlated with that of their caregivers compared to free-ranging dogs. Furthermore, activity of kennel-housed sled dogs was primarily driven by human activity rather than environmental variables [6]. Many studies have also explored the relationship between dog and human sleep patterns through actigraphy measures [7–9]. Actigraphy studies on human-dog dyads that co-sleep (sleep in the same bed or bedroom) have shown dog activity to have a greater influence on human activity rather than human activity on dog activity [7, 8]. For example, Hoffman and colleagues [8] found that dog movement transitioning to human movement was more likely than human movement transitioning to dog movement, which suggests that human wake-ups are more dependent on dog activity. During nighttime, dogs were awake approximately three times longer than their caregivers, which may be attributed to their shorter sleep-wake cycles [8]. Taken together, these studies show that dog activity has a greater influence on human activity than vice versa, and that dog activity shows greater flexibility due to their short sleep cycles [10].

The aim of this study is to assess how readily dogs adjust to the Fall Back DST change in Canada. Given the high entrainment of dog and human activities, we hypothesize that dogs' adjustment to DST will mirror their human counterparts. Specifically, we expect sled dogs, with their rigid schedules, to show significant shifts in morning activity patterns immediately after DST, similar to their human handlers. In contrast, companion dogs, with more flexible routines influenced by their caregivers' schedules, are predicted to exhibit variable adjustment patterns. Additionally, we hypothesize that dogs with highly regulated routines will show a more pronounced alignment in their adjustment to DST. Therefore, sled dogs are anticipated to adjust more immediately and consistently compared to companion dogs. Alternatively, dogs may not show a consistent adjustment to DST, independent of their human counterparts. This would suggest no significant difference in adjustment patterns between sled dogs and companion dogs, indicating that factors beyond human-imposed schedules, such as individual temperament or environmental cues, are critical. Under this scenario, we expect high variation in dogs' adjustment to DST, with no clear pattern of alignment with their human counterparts.

## Methods

### Data collection

This study includes two dog groups: sled dogs from Haliburton Forest & Wild Life Reserve in Haliburton, Ontario, Canada (45.22˚ N, 78.59˚ W); and companion dog and human caregiver dyads from Ontario and Quebec, Canada. Data were collected from 25 sled dogs (S1 Table) from October 25, 2020 to December 3, 2020 (clocks turned back one hour at 2:00 am on November 1, 2020). For background information on kennel setup, feeding, and activity schedules at Haliburton Forest & Wild Life Reserve, see S1 File.

For companion dog and caregiver dyads, human participants over the age of 18 years who lived in or near Ontario and who had either a husky or malamute dog were recruited for this study through poster advertising on social media and through word-of-mouth referrals. Husky and malamutes were chosen since these breeds are traced back to an ancient lineage of dogs originating from Zokhov Island, Siberia associated with sledding [11]. We also wanted to control for any breed-level differences in activity patterns [12]. Dogs over the age of 13 or who had prior medical issues that would influence their activity patterns were excluded from the study. Interested individuals who met the criteria and could commit to the study requirements took part in the study with their dogs. Data were collected from 37 companion dogs from November 2, 2021 to December 9, 2021 (clocks turned back one hour at 2:00 am on November 7, 2021).

One sled dog and eight companion dogs (and their associated human caregiver) were excluded from all analyses (see S1 Table). CAN017, CAN095 lost their sensors during the data collection period; the associated caregiver of CAN095, CAH037 was also excluded from the analyses. CAN065 and dyad CAN072-CAH024 were excluded due to incomplete data, meaning the sensor did not pick up activity for part of the data collection period. Five companion dog-caregiver dyads (CAN061-CAH016, CAN062-CAH017, CAN066-CAH020, CAN082-CAH032, CAN098-CAH040) were excluded because they did not have enough pretransition data (less than two days). Including individuals with less than two pre-transition dates would potentially skew data as a result of the weekend effect [13, 14]. Dyad CAN087-CAH034 were excluded since the caregiver did not make any changes to routines for DST. Our results are not expected to be affected by the exclusion.

The CamNtech MotionWatch 8 accelerometer was used to record human and dog activity. The accelerometer records and quantifies movement as acceleration waveforms per second [15]. We set the accelerometer to record activity for every 1-minute epoch (summation of MotionWatch counts over a 1-minute interval). For dogs, the accelerometer was fastened to a nylon collar with Gorilla Tape similar to the attachment protocol used for sled dogs in [6]. Human participants wore the accelerometer around their wrist and were instructed to wear it on their non-dominant hand. Once materials were prepped, the items were shipped to each participant. The recruitment process and interactions were done remotely as a result of the provincial lockdown measures placed due to the COVID-19 pandemic. Once materials were delivered, human participants wore the watch and placed collars on dogs and noted the date and time. Activity was recorded until the end of the data collection period at which point, caregivers removed the collars and watches and shipped the items back to researchers. The raw activity count was then downloaded using MotionWare software (version 1.2.23). Human participants provided demographic information and details about their schedule and relationship with their dogs as a part of a larger study. We obtained the sunrise and sunset times for each city from sunrise-sunset.org.

To determine human sleep measures, author MFL first cleaned the raw actigraphy data by deleting periods of non-wear times (i.e., when watch was taken off) that exceeded 30 minutes.

We had asked participants to press an Event marker button to signal when they went to bed and got up and these markers were used to assist with scoring sleep periods. MFL visually determined the period between when participants went to bed (decrease in activity levels) and the "Got-up Time" which was when participants got out of bed (increase and consistency in activity levels). MotionWare then algorithmically categorized each 1-minute epoch as "sleep" or "wake" throughout the sleep period—this is how MotionWare designates "Woke-up Time" (sleep offset: when participants woke up) and this time always preceded the scored "Got-up Time" [15].

## Ethics statement

Ethical approval was obtained for this study through the University of Toronto and written consent was obtained from all participating parties prior to data collection. Methods used for this study are in accordance with the guidelines and regulations in place by the University of Toronto Animal Care Committee (Protocol # 20012651) and the University of Toronto Human Research Ethics Board (Protocol # 40107). Methods were also reviewed and approved by staff at Haliburton Forest & Wild Life Reserve. This study complies with the Animal Research: Reporting of In Vivo Experiments (ARRIVE) guidelines [16]. Human participants received monetary compensation for their participation.

## Data analyses

**Sled dogs.**   To determine how long it took sled dogs to adjust to DST transition, we looked at two activity measures: morning onset activity according to handler arrival time (sum of activity between 6:31 am and 7:30 am, which was one hour before handlers arrived at 7:30 am), and morning onset activity according to sunrise time (sum of activity 30 minutes before and after sunrise). To calculate sled dogs' activity measures pre-transition, we averaged activity measures across the seven days before DST transition (October 25 to October 31). We then compared the activity measure for each of the three consecutive days following DST transition to the pre-transition average (Pre). In other words, we compared DST1 (November 1) to Pre, DST2 (November 2) to Pre, and DST3 (November 3) to Pre. Both morning onset activity measures were not normally distributed so we used Wilcoxon signed rank tests. We also used a one-way repeated measures ANOVA to determine whether there were differences among the three post-transition dates. Again, since the morning onset activities were not normally distributed, we used the Friedman test. See S1 File for additional analyses the effect of DST on total daily activity (sum of activity over 24-hour period; S1 Fig) and the effects of sex and age on morning onset activities for sled dogs (S2 and S3 Tables).

**Companion dogs.**   To determine how long it took companion dogs to adjust to DST transition, we looked at two activity measures: morning onset activity according to their caregiver's Got-up Time (sum of activity 30 minutes before and after caregivers' Got-up Time, as scored via MotionWare), and morning onset activity according to sunrise time (sum of activity 30 minutes before and after sunrise). To calculate companion dogs' activity measures pre-transition, we averaged activity measures across the days before DST transition. Days pre-transition varied (ranged from 2 to 5 days; November 2 to November 6) depending on when participants received the study equipment and data collection began. We excluded five dyads (S1 Table) that only had one day of data pre-transition since this would have been a Saturday and dog activity is known to differ on weekends compared to weekdays [e.g., 15, 16]. Similar to the sled dog analyses, we then compared the activity measure for each of the three consecutive days following DST transition to the pre-transition measure (Pre). In other words, we compared DST1 (November 7) to Pre, DST2 (November 8) to Pre, and DST3 (November 9) to Pre. Both morning onset activity measures were not normally distributed so we used Wilcoxon signed

rank tests. We also used the Friedman test to see whether there were differences among the three post-transition dates. See S1 File for additional analyses on total daily activity (S2 Fig) and the effects of sex, age, and presence of other dogs in the household on morning onset activities for companion dogs (S2 and S3 Tables).

**Human caregivers.** To determine how DST transition affected human sleep and activity, we compared caregiver Woke-up Time and Got-up Time for each of the three consecutive days following DST transition to the pre-transition measure (days were the same as for companion dogs). Both Woke-up Time and Got-up Time were normally distributed so we used paired t-tests. We also used a one-way repeated measures ANOVA to determine whether there were differences among the three post-transition dates.

All statistical analyses were performed in R Studio (v.2023.12.1+402; R v.4.0.1) for Mac OS X [17]. All statistical tests were two-tailed, with alpha set to 0.05 for significance.

## Results

For sled dogs, mean morning onset activity according to handler arrival time was 3659.02 (± SD 2137.88) Pre-DST and 10983.04 (± SD 6026.74) on DST1. Mean morning onset activity according to sunrise time was 23844.95 (± SD 12596.34) Pre-DST and 7603.75 (± SD 5055.11) on DST1. For companion dogs, mean morning onset activity according to caregiver got-up time was 11951.21 (± SD 14310.27) Pre-DST and 12031.07 (± SD 16653.6) on DST 1. Mean morning onset activity according to sunrise time was 15023.58 (± SD 23189.57) Pre-DST and 5060.655 (± SD 5158.09) on DST1. See Table 1 for the complete descriptive statistics of all dog and human activity and sleep measures pre- and post-DST transition.

### Sled dogs

For morning onset activity based on handler arrival time, we found that only DST1 differed significantly from pre-transition (Table 2, Fig 1A and 1B). Dogs were significantly more active on DST1 than pre-transition. The Friedman test showed a significant difference among the three post-transition dates ($\chi^2 = 25.33$, $p < 0.001$). Nemenyi post hoc pairwise comparisons (with Bonferroni corrections) showed that dogs were significantly more active on DST1 than DST2 ($p < 0.001$) and DST3 ($p < 0.001$). However, there was no significant difference between DST2 and DST3.

For morning onset activity based on sunrise time, we found that all three post-transition dates differed significantly from pre-transition (Table 2, Fig 1C and 1D). Dogs were less active in all three days following transition compared to pre-transition. The Friedman test showed a significant difference among the three post-transition dates ($\chi^2 = 15.08$, $p < 0.001$). Nemenyi post hoc tests (with Bonferroni corrections) showed that dogs were significantly more active on DST1 than DST2 ($p = 0.017$) and DST3 ($p < 0.001$). However, there was no significant difference between DST2 and DST3.

**Table 1. Descriptive statistics of the mean (standard deviation) for morning onset activities in dogs and Woke-up and Got-up Times in human caregivers.**

|  | Sled Dogs | | Companion Dogs | | Human Caregivers | |
|---|---|---|---|---|---|---|
| Date | Morning activity—handler | Morning activity—sunrise | Morning activity—caregiver | Morning activity—sunrise | Woke-up Time (hh:mm) | Got-up Time (hh:mm) |
| Pre-DST | 3659.02 (2137.88) | 23844.95 (12596.34) | 11951.21 (14310.27) | 15023.58 (23189.57) | 7:46 am (1:11) | 7:52 am (1:11) |
| DST1 | 10983.04 (6026.74) | 7603.75 (5055.11) | 12031.07 (16653.6) | 5060.655 (5158.09) | 8:03 am (2:20) | 8:10 am (2:22) |
| DST2 | 4123.88 (5442.31) | 4577.13 (5637.06) | 12591.38 (16088.2) | 15294.93 (27702.82) | 6:50 am (1:11) | 6:54 am (1:12) |
| DST3 | 4197.88 (4560.79) | 4207.29 (4522.72) | 16474.79 (30832.71) | 25003.52 (58877.07) | 7:09 am (1:14) | 7:17 am (1:14) |

**Table 2. Results from pairwise comparisons of pre- and post-DST transition activity measures in sled dogs and companion dogs.**

| | Sled Dog Analysis | | Companion Dog Analysis | |
|---|---|---|---|---|
| Comparison | Morning onset (handler) | Morning onset (sunrise) | Morning onset (caregiver) | Morning onset (sunrise) |
| Pre–DST1 | ***p* < 0.001** | ***p* < 0.001** | *p* = 0.508 | *p* = 0.053 |
| Pre–DST2 | *p* = 0.527 | ***p* < 0.001** | *p* = 0.831 | *p* = 0.624 |
| Pre–DST3 | *p* = 0.966 | ***p* < 0.001** | *p* = 0.865 | *p* = 0.624 |

Significant effects ($p < 0.05$) are bolded.

## Companion dogs

For morning onset activity based on caregiver get-up times, we found no difference between pre-transition and the three post-transition dates (Table 2; Fig 2A and 2B). The Friedman test did not show any significant difference among the three post-transition dates ($\chi^2 = 0.62$,

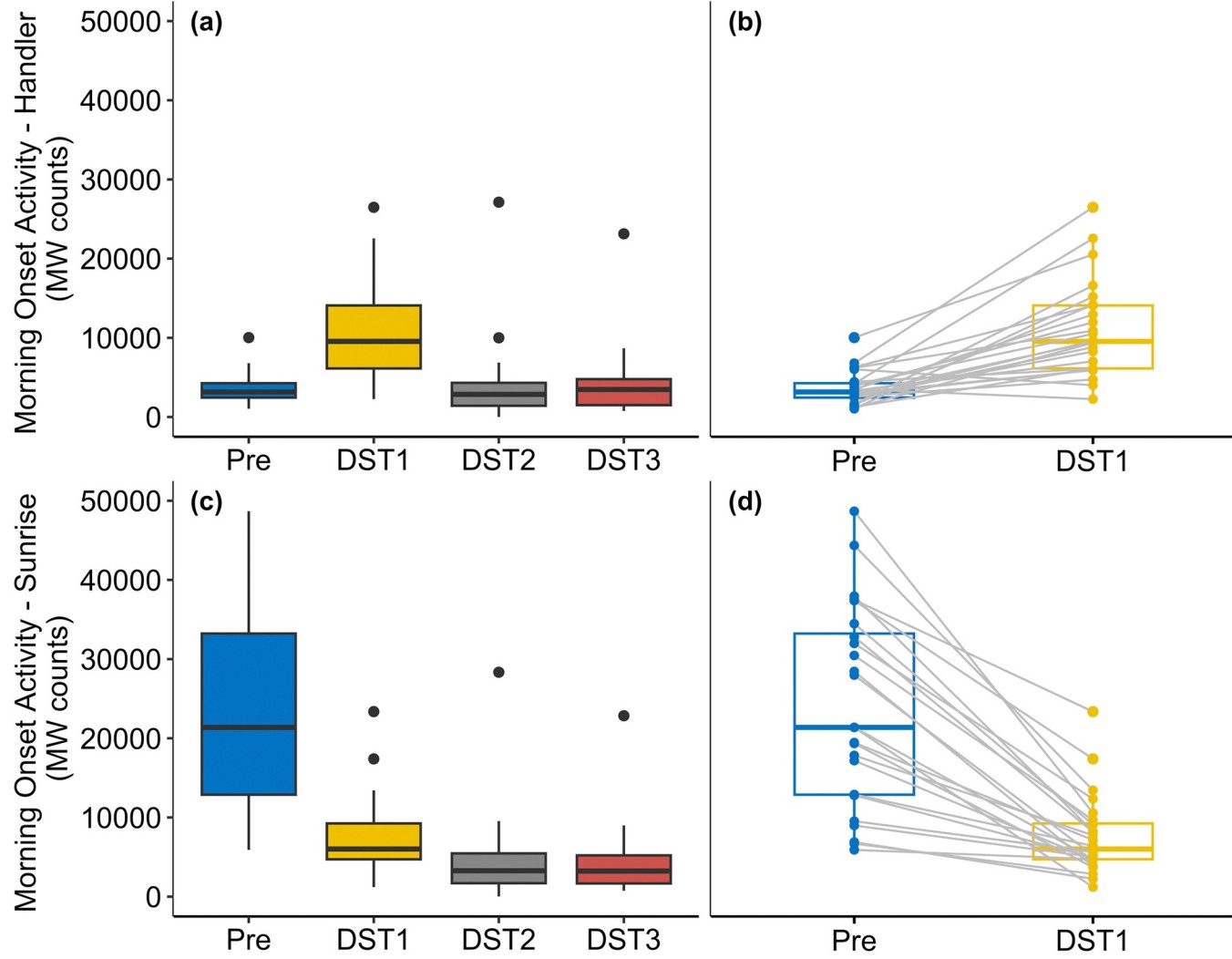

**Fig 1.** Box plots comparing sled dog morning onset activity based on handler arrival time (a, b) and sunrise time (c, d) for pre- and post-DST transition. Grey lines in the paired box plots (b, d) denote within-individual differences between pre-DST and the day of DST.

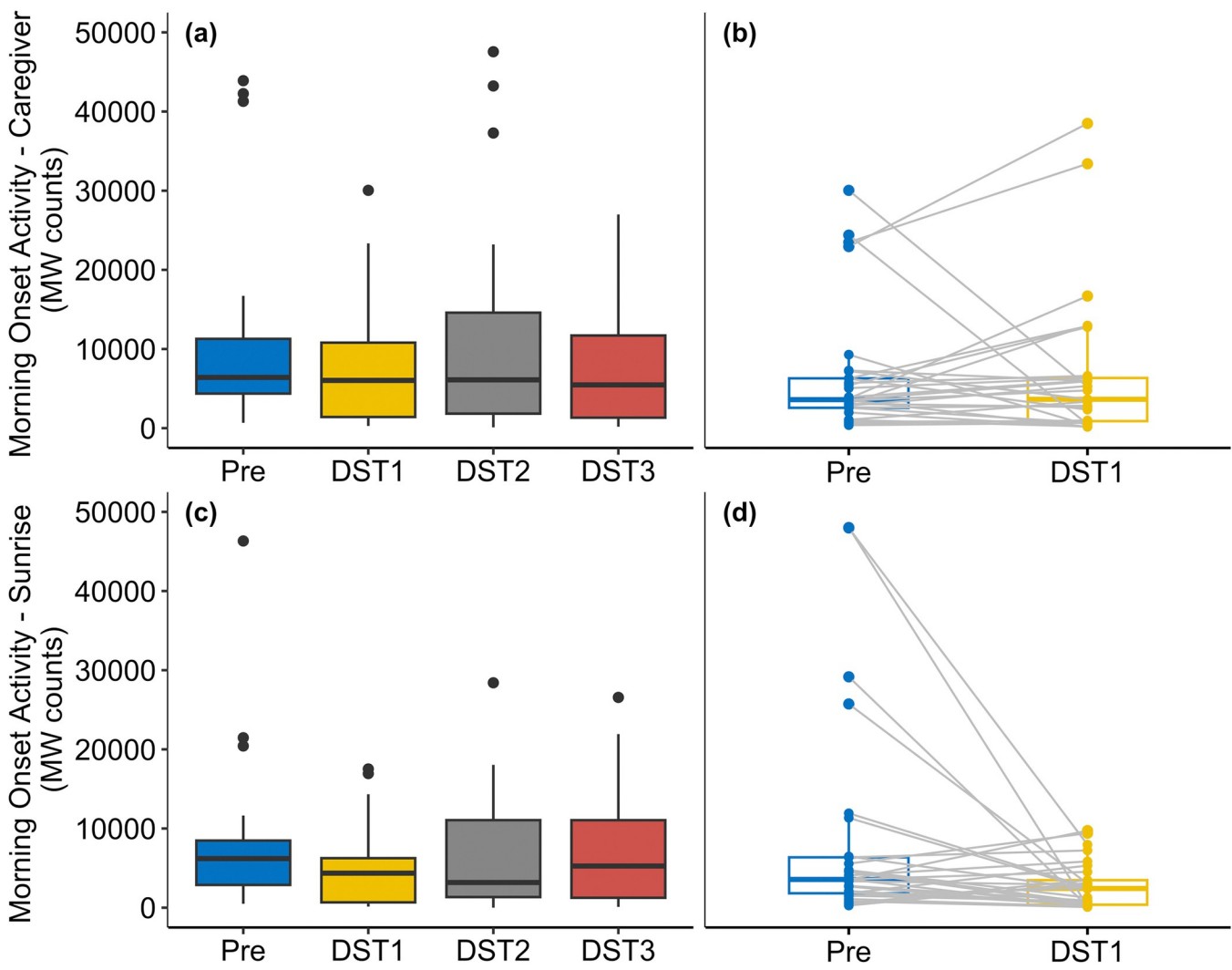

**Fig 2.** Box plots comparing companion dog morning onset activity based on caregiver Got-up Time (a, b) and sunrise time (c, d) for pre- and post-DST transition. Grey lines in the paired box plots (b, d) denote within-individual differences between pre-DST and the day of DST.

$p = 0.733$). Similarly, for morning onset activity based on sunrise time, we found no difference between pre-transition and the three post-transition dates (Table 2; Fig 2C and 2D). The Friedman test did not show any significant difference among the three post-transition dates ($\chi^2 = 3.38$, $p = 0.185$).

## Humans

We found no significant difference in caregiver Woke-up Time between DST1 and Pre ($p = 0.585$), however, Woke-up Times on DST2 ($p < 0.001$) and DST3 ($p = 0.046$) were significantly earlier than Pre (Fig 3A and 3B). We found no significant difference in Got-up Time between DST1 and Pre ($p = 0.542$) and between DST3 and Pre ($p = 0.054$); see Fig 3C and 3D. Got-up Times on DST2 ($p < 0.001$) was significantly earlier than Pre. The ANOVA indicated a significant difference among three post-transition dates for Woke-up Time (F = 7.49, $p = 0.018$) and Got-up Time (F = 8.03, $p = 0.015$). However, Wilcox post hoc pairwise

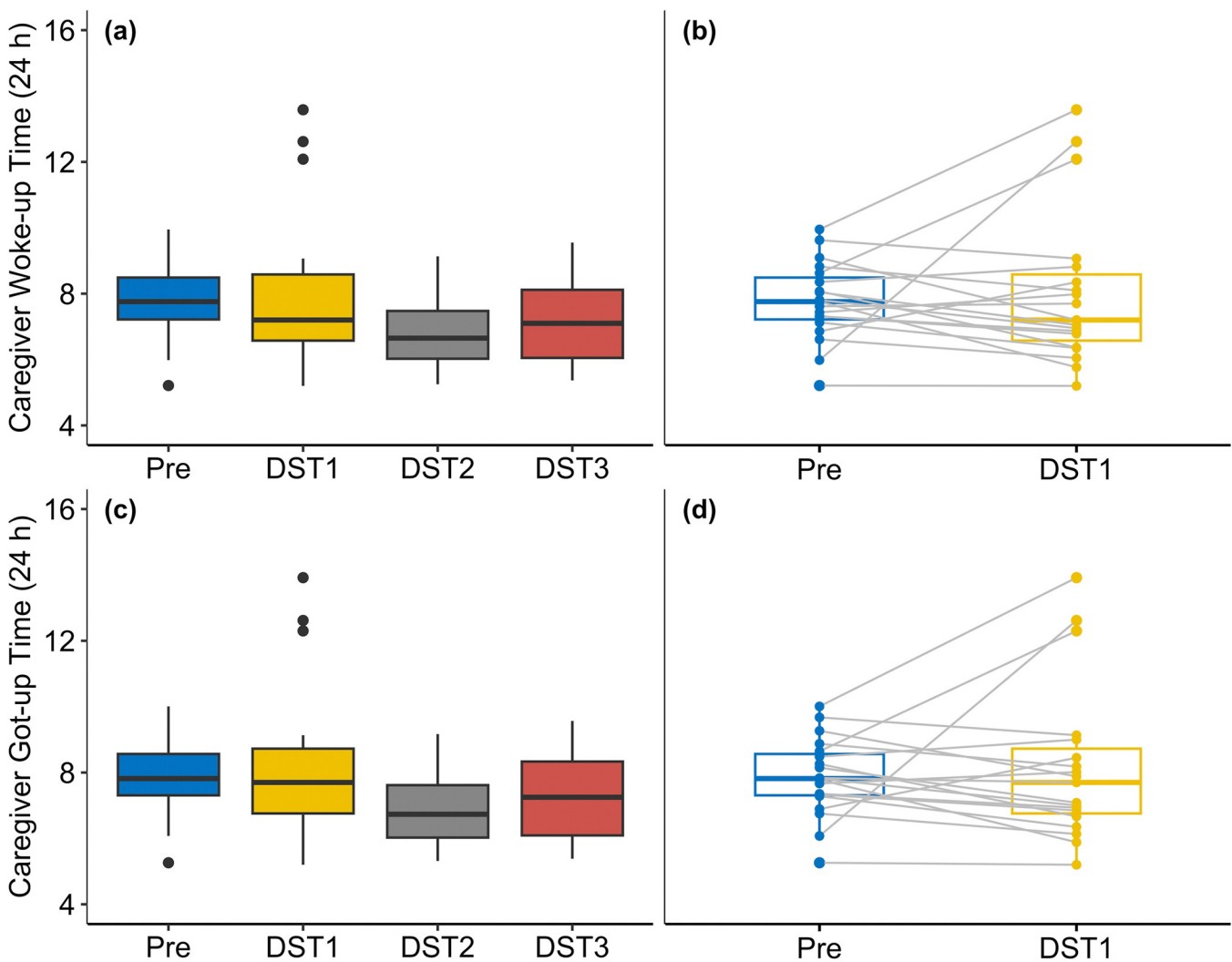

**Fig 3.** Box plots comparing caregiver Woke-up Time (a, b) and Got-Up Time (c, d) for pre- and post-DST transition. Grey lines in the paired box plots (b, d) denote within-individual differences between pre-DST and the day of DST.

comparisons (with Bonferroni corrections) did not show significant differences between any of the DST dates for Woke-up and Got-up Times.

### Effect of age on dog morning activity

In the models examining the effects of age, sex, and presence of other dogs on dogs' morning onset activity, the only significant effect we found was that age was negatively associated with sled dogs' morning onset activity based on handler arrival time. Older sled dogs tended to be less active when handlers came into the kennel in the morning ($\beta$ = -0.113, $p$ = 0.012). In the models looking at the change in morning onset activity on DST1, the only significant effect we found was that age was negatively associated with the difference between companion dogs' DST1 and Pre-DST morning onset activity based on caregiver got-up time. Older companion dogs had decreased morning activity on DST1 compared to Pre-DST ($\beta$ = -2249.8, $p$ = 0.028). Full model results are reported in S2 and S3 Tables.

## Discussion

Here, we report how sled dogs and companion dogs in Canada adjusted to the Fall Back transition. In sled dogs, we found their morning onset activity based on handler arrival time to be significantly higher on DST Sunday compared to pre-transition, and resumed to pre-transition levels on DST2 and DST3. In other words, sled dogs took one day to adjust to the time shift. In companion dogs, we did not find evidence for any changes in morning onset activity following DST. Similarly, we did not find caregivers' Woke-up and Got-up Times to change on DST Sunday. However, we did find that while caregivers woke up earlier on the weekdays after DST, dogs maintained their pre-transition morning activity levels. Morning activity in both groups of dogs did not appear to follow sunrise times. Each of these findings will be discussed below.

### Effect of sunrise on dog morning activity

Companion dogs showed no significant difference in their morning activity around sunrise time pre- compared to post-DST transition. Survey responses indicate that most companion dogs slept indoors (one dog consistently slept outdoors, and five dogs occasionally slept outdoors). Conversely, in sled dogs, their morning onset activity based on sunrise time was markedly higher pre-DST transition compared to all three days after DST. Sunrise time pre-DST (~7:50 am) had greater overlap with handler arrival time (7:30 am) compared to sunrise time immediately following DST (~ 6:50 am)—dogs remained inactive around sunrise after DST since handler arrival time was much later than sunrise. It is important to note that activity was higher on the day of DST relative to DST2 and DST3 which reflects sled dogs taking one day to adjust to the delay in handler arrival time. In diurnal mammals, sleep-wake patterns are entrained to the light-dark cycle due to melatonin, the main sleep-promoting hormone, being produced during the dark period of the day [4]. Recently, it has been shown that endogenous cues, such as light, affect activity patterns in dogs under laboratory conditions [18, 19]. However, many studies have shown that the activity of companion dogs strongly aligns with human routines [13, 20, 21], perhaps more than endogenous environmental variables [6]. In our study, we found that companion dogs' morning activity was not entrained to sunrise, and this was also the case in outdoor kennel-housed sled dogs whose activity adjusted according to handler arrival time rather than sunlight.

### Sled dog activity shift

Sled dogs follow very rigid morning routines since handler arrival time is always 7:30 am (clock time). As such, sled dogs have daily schedules that are more regulated by humans than the companion dogs in our study who faced flexible caregiver work routines and the presence of multiple caregivers in the house (discussed below). Drug detector dogs that returned to work after an extended period of leave took one day to adjust ("first-day-back effect") [10], which was what we observed in sled dogs adjusting to DST. Flexibility in activity patterns may be attributed to shorter sleep-wake cycles [10, 22]. We suspect there may also be a synchrony effect since sled dogs are all housed in the same sex-specific enclosure so when one individual exhibits morning onset activity, the group may follow suit. Adams and Johnson [23] found that group-housed dogs were able to differentiate between different auditory stimuli even while they were asleep. Dogs were more likely to respond to other dogs' barking compared to anthropogenic noises. Circadian rhythm synchrony was found in cohabiting marmoset pairs [24] and mice [25]. Future studies should quantitatively assess activity synchrony among group-housed dogs.

## Human activity and companion dogs

Our results indicate that DST did not affect Woke-up and Got-up Times in humans, therefore, it is not surprising that DST did not affect morning onset activity in companion dogs. It also does not appear that dogs' routines are dictating human wake-up times (i.e., neither humans nor dogs woke up 1 hour earlier on DST Sunday to adjust for morning needs). Caregiver Woke-up Times were earlier on DST2 and DST3 than before the DST change. DST2 followed the same pattern in caregiver Got-up Times but DST 3 did not show an increase in activity compared to pre-transition dates. Since DST2 and DST3 were Monday and Tuesday, respectively, the pattern in Woke-up Times may be indicative of caregivers waking up earlier on workdays compared to the previous Sunday (DST1). Dogs, on the other hand, did not show this pattern of increased morning onset activity during weekdays after DST. Social jetlag [26], defined as the mismatch between the internal biological clock and the social time one has to adhere to (e.g., work schedule), has been found to be stronger in human caregivers than dogs [21].

The lack of tight coupling between dog and caregiver morning activity following DST could be due to several study limitations. First, dogs slept in varying proximity to their human caregiver, thus, sleeping distance could impact the likelihood of human disruptions on dog activity and dog disruptions on human activity [8]. Second, we did not take into consideration the influence of other caregivers in the household that might affect dog morning activity. Third, in this study group, there was considerable variability in caregivers' work arrangements, especially in light of the COVID-19 pandemic, which required flexibility in work accommodations (working in-office, fully remote, and hybrid). Regardless, survey responses indicated that dogs were spending the majority of their time with humans irrespective of the primary caregiver's working commitments (i.e., more than half of the dogs spent less than 5 hours alone daily), therefore dogs' morning activity could have been entrained to other members of the household.

The findings from our study reflects dog response to the Fall Back DST change in or near Ontario, Canada and may not mirror other regions that have different dog keeping practices. Companion dogs in our study showed variability in sleeping location, feeding times, and daily schedules (e.g., walk times). In this field study, we did not alter the daily routines of dogs to collect their daily activity data; therefore, we could not control for environmental or situational factors such as extreme weather conditions or changes made to day-to-day schedules. Future research could implement a daily "activity log" for caregivers to note detailed information about their daily schedule.

## Effects of age, sex, and presence of other dogs on morning activity

We found that older sled dogs generally had lower morning activity levels around handler arrival time than younger dogs. In addition, older companion dogs had lower activity on DST Sunday compared to pre-DST (S1 File)—this finding was unexpected and warrants further research that specifically looks at how older dogs respond to schedule changes. We recommend caregivers of older dogs to be especially mindful in implementing sudden changes to daily routines. The general trend of dog activity decreasing with age concurs with findings from other studies [e.g., 5, 16]. However, in a previous study with the same group of sled dogs, we did not find age to have an effect on total daytime activity [6]. Perhaps older dogs have decreased morning activity levels, or older dogs may be less reactive to human presence, compared to younger dogs. Interestingly, neither sex nor presence of other dogs in the household (i.e., among companion dogs) affected morning activity levels or the change in morning activity level following DST.

## Conclusion

The significance of this research lies in its illumination of the intricate and bidirectional relationship between humans and their canine companions. This study underscores how human-imposed schedule changes, such as DST, can ripple through the daily lives of dogs, affecting their activity patterns and well-being. Understanding these effects is crucial for both dog owners and professionals working with dogs, as it can inform strategies to mitigate potential stressors associated with time changes. While the majority of literature examining the effect of DST on sleep and well-being has focused on humans, our paper is the first to quantify how this sudden, human-mitigated change affects the routines of domestic dogs. By recognizing the interconnectedness of human and dog activities, we can better appreciate the complexities of our shared environments and work towards fostering living conditions that support the health and well-being of both humans and their canine companions.

## Supporting information

**S1 Table. Summary of information for study participants.** Associated human ID is provided for companion dogs to note dyads. Information about breed, sex, age and weight is provided for canids. Age in years and weight in kilograms reflects that at time of data collection. Asterisks (*) next to ID notes individuals that were excluded for the study, see note below chart for more detailed reasoning.
(DOCX)

**S2 Table. Results from linear mixed-effects models on the effects of sex, age, and other dogs on morning onset activities for sled dogs and companion dogs.**
(DOCX)

**S3 Table. Results from linear regressions on the effects of sex, age, and other dogs on morning onset activity difference between DST1 and pre-DST for sled dogs and companion dogs.**
(DOCX)

**S1 Fig. Box plots comparing sled dog total daily activity for pre- and post-DST transition dates in sled dogs.**
(TIF)

**S2 Fig. Box plots comparing companion dog total daily activity for pre- and post-DST transition dates in companion dogs.**
(TIF)

**S1 File. Additional information on sled dogs, analyses on total daily activity, and analyses on the effects of age, sex, and presence of other dogs on morning activity.**
(DOCX)

**S2 File. Dataset used to generate study results.**
(XLSX)

## Acknowledgments

We are grateful for the participation of both dogs and caregivers for providing data for this study. This research would not be possible without the involvement of caregivers and staff at Haliburton Forest & Wild Life Reserve who provided background information and had the responsibility of equipment attachment.

## Author Contributions

**Conceptualization:** Lavania Nagendran, Ming Fei Li, David R. Samson, Lauren Schroeder.

**Data curation:** Lavania Nagendran, Ming Fei Li.

**Formal analysis:** Ming Fei Li.

**Funding acquisition:** David R. Samson, Lauren Schroeder.

**Methodology:** Lavania Nagendran, Ming Fei Li.

**Resources:** Lavania Nagendran, Ming Fei Li.

**Supervision:** David R. Samson, Lauren Schroeder.

**Visualization:** Ming Fei Li.

**Writing – original draft:** Lavania Nagendran, Ming Fei Li, David R. Samson, Lauren Schroeder.

**Writing – review & editing:** Lavania Nagendran, Ming Fei Li, David R. Samson, Lauren Schroeder.

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
