## [Decision Letter · Decision Letter 0]

11 Nov 2024

PONE-D-24-30341The impact of Daylight SavingTime on dog activityPLOS ONE

Dear Dr. Nagendran,

Thank you for submitting your manuscript to PLOS ONE. After careful consideration, we feel that it has merit but does not fully meet PLOS ONE’s publication criteria as it currently stands. Therefore, we invite you to submit a revised version of the manuscript that addresses the points raised during the review process.

We look forward to receiving your revised manuscript.

Kind regards,

Vinícius Silva Belo

Academic Editor

PLOS ONE

Journal Requirements:

“his research was supported through Discovery Grants from the Natural Sciences and Engineering Research Council of Canada (RGPIN-2020-04159 to L.S. and RGPIN-2020-05942 to D.R.S).”

4. In this instance it seems there may be acceptable restrictions in place that prevent the public sharing of your minimal data. However, in line with our goal of ensuring long-term data availability to all interested researchers, PLOS’ Data Policy states that authors cannot be the sole named individuals responsible for ensuring data access (http://journals.plos.org/plosone/s/data-availability#loc-acceptable-data-sharing-methods).

5. We note that there is identifying data in the Supporting Information file <Supporting Information.docx>. Due to the inclusion of these potentially identifying data, we have removed this file from your file inventory. Prior to sharing human research participant data, authors should consult with an ethics committee to ensure data are shared in accordance with participant consent and all applicable local laws.

-Location data

Please remove or anonymize all personal information (location), ensure that the data shared are in accordance with participant consent, and re-upload a fully anonymized data set. Please note that spreadsheet columns with personal information must be removed and not hidden as all hidden columns will appear in the published file.

**Additional Editor Comments:**

In addition to the comments provided by the reviewer, the following points should be considered:

-The reasons for the exclusion of 8 dogs should be stated in the text and in more detail than in the supplementary material. In addition, possible effects of the exclusions should be discussed.

-The text should present information that can be highlighted from Table 1.

Reviewers' comments:

Reviewer's Responses to Questions

**Comments to the Author**

1. Is the manuscript technically sound, and do the data support the conclusions?

Reviewer #1: Yes

2. Has the statistical analysis been performed appropriately and rigorously? 

Reviewer #1: Yes

3. Have the authors made all data underlying the findings in their manuscript fully available?

Reviewer #1: Yes

4. Is the manuscript presented in an intelligible fashion and written in standard English?

Reviewer #1: Yes

5. Review Comments to the Author

Reviewer #1: The study addresses a relevant topic: the influence of time changes, such as Daylight Saving Time, on dogs' routines, which can have significant implications for the animals' well-being and the routine of their guardian. The research used accelerometers to measure the dogs' activity, providing objective and quantitative data on their behavior during the time transition. If data are available, I suggest (but not as a requirement) that association tests be carried out between the change in behavior and some variables, such as: Dogs' age: Including an analysis by age can be a valuable addition, since different age groups have different activity levels and behavior patterns. Presence of other animals in the guardian's home: This analysis would allow us to observe whether interaction with other animals influences canine behavior during the time change. Where the animal is raised (indoors or outdoors): Considering this variable may be relevant, since living conditions (whether the animal lives indoors or outdoors) can impact both activity patterns and the dogs' adaptation to time changes. The inclusion of these variables could provide a broader view of the factors that influence dogs' behavior when faced with time changes, increasing the value and practical applicability of the research results.

6. PLOS authors have the option to publish the peer review history of their article (what does this mean?). If published, this will include your full peer review and any attached files.

Reviewer #1: No

---

## [Author Response · Author response to Decision Letter 0]

1 Dec 2024

Thank you for the helpful comments. Please see the 'Response to Reviewers' document for responses in red. Pasted below are editor and review comments with authors' responses. 

Journal Requirements:

We will rename all files according to PLOS ONE’s style requirements.

“his research was supported through Discovery Grants from the Natural Sciences and Engineering Research Council of Canada (RGPIN-2020-04159 to L.S. and RGPIN-2020-05942 to D.R.S).”

We have added this amended Role of Funder statement at the bottom of our revision cover letter.

We decided to upload the raw data used to generate the results of our study as a “Supporting Information” file. We have updated our Data Availability statement in the submission form.

4. In this instance it seems there may be acceptable restrictions in place that prevent the public sharing of your minimal data. However, in line with our goal of ensuring long-term data availability to all interested researchers, PLOS’ Data Policy states that authors cannot be the sole named individuals responsible for ensuring data access (http://journals.plos.org/plosone/s/data-availability#loc-acceptable-data-sharing-methods).

Data was uploaded as a Supporting Information file. Please see the above comment.

5. We note that there is identifying data in the Supporting Information file <Supporting Information.docx>. Due to the inclusion of these potentially identifying data, we have removed this file from your file inventory. Prior to sharing human research participant data, authors should consult with an ethics committee to ensure data are shared in accordance with participant consent and all applicable local laws.

-Location data

Please remove or anonymize all personal information (location), ensure that the data shared are in accordance with participant consent, and re-upload a fully anonymized data set. Please note that spreadsheet columns with personal information must be removed and not hidden as all hidden columns will appear in the published file.

We have removed “Location” from the Supporting Information file.

We have reviewed our reference list and ensured that it is complete and correct. We did not add or remove any references.

Additional Editor Comments:

In addition to the comments provided by the reviewer, the following points should be considered:

-The reasons for the exclusion of 8 dogs should be stated in the text and in more detail than in the supplementary material. In addition, possible effects of the exclusions should be discussed.

We included the reason for the exclusion of 1 sled dog and 8 companion dogs, and that we do not expect these exclusions to change our study findings. This information was added to the manuscript text (lines 105-115). 

-The text should present information that can be highlighted from Table 1.

We have added text to the main manuscript that highlights pertinent information from Table 1 (lines 195-201).

Reviewers' comments:

Reviewer's Responses to Questions

Comments to the Author

1. Is the manuscript technically sound, and do the data support the conclusions?

Reviewer #1: Yes

2. Has the statistical analysis been performed appropriately and rigorously?

Reviewer #1: Yes

3. Have the authors made all data underlying the findings in their manuscript fully available?

Reviewer #1: Yes

4. Is the manuscript presented in an intelligible fashion and written in standard English?

Reviewer #1: Yes

5. Review Comments to the Author

Reviewer #1: The study addresses a relevant topic: the influence of time changes, such as Daylight Saving Time, on dogs' routines, which can have significant implications for the animals' well-being and the routine of their guardian. The research used accelerometers to measure the dogs' activity, providing objective and quantitative data on their behavior during the time transition. If data are available, I suggest (but not as a requirement) that association tests be carried out between the change in behavior and some variables, such as: Dogs' age: Including an analysis by age can be a valuable addition, since different age groups have different activity levels and behavior patterns. Presence of other animals in the guardian's home: This analysis would allow us to observe whether interaction with other animals influences canine behavior during the time change. Where the animal is raised (indoors or outdoors): Considering this variable may be relevant, since living conditions (whether the animal lives indoors or outdoors) can impact both activity patterns and the dogs' adaptation to time changes. The inclusion of these variables could provide a broader view of the factors that influence dogs' behavior when faced with time changes, increasing the value and practical applicability of the research results.

We thank the reviewer for these helpful suggestions. We agree that these additional analyses would provide further insight on dogs’ morning behavior and how they may respond to time changes. For sled dogs, we included additional analyses on how sex and age affect general morning activity levels and the change in morning activity on DST1 compared to Pre-DST. We did not include the presence of other dogs since all sled dogs live in kennels with other dogs and all kennels are located close to each other and are outdoors. For companion dogs, we included additional analyses on how sex, age, and presence of other dogs in the household affect general morning activity levels and the change in morning activity on DST1 compared to Pre-DST. We did not include location since all companion dogs, except for four, slept indoors during the study period. The four dogs that sometimes slept outside were all from the same household. Our main findings were that, compared to younger dogs, older sled dogs had lower morning onset activity around handler arrival time, and older companion dogs had lower morning onset activity based on caregiver got-up time on DST1 compared to Pre-DST.

We included the additional analyses and results in the Supporting Information file as well as in the main manuscript (lines 165-166, 183-184, 251-260). We also added an additional section to our Discussion interpreting results from these additional analyses (lines 336-348).

6. PLOS authors have the option to publish the peer review history of their article (what does this mean?). If published, this will include your full peer review and any attached files.

Do you want your identity to be public for this peer review? For information about this choice, including consent withdrawal, please see our Privacy Policy.

Reviewer #1: No

---

## [Decision Letter · Decision Letter 1]

20 Dec 2024

The impact of Daylight SavingTime on dog activity

PONE-D-24-30341R1

Dear Dr. Nagendran,

We’re pleased to inform you that your manuscript has been judged scientifically suitable for publication and will be formally accepted for publication once it meets all outstanding technical requirements.

Kind regards,

Vinícius Silva Belo

Academic Editor

PLOS ONE

Additional Editor Comments (optional):

Reviewers' comments:

Reviewer's Responses to Questions

**Comments to the Author**

1. If the authors have adequately addressed your comments raised in a previous round of review and you feel that this manuscript is now acceptable for publication, you may indicate that here to bypass the “Comments to the Author” section, enter your conflict of interest statement in the “Confidential to Editor” section, and submit your "Accept" recommendation.

Reviewer #1: All comments have been addressed

2. Is the manuscript technically sound, and do the data support the conclusions?

Reviewer #1: Yes

3. Has the statistical analysis been performed appropriately and rigorously? 

Reviewer #1: Yes

4. Have the authors made all data underlying the findings in their manuscript fully available?

Reviewer #1: Yes

5. Is the manuscript presented in an intelligible fashion and written in standard English?

Reviewer #1: Yes

6. Review Comments to the Author

Reviewer #1: (No Response)

7. PLOS authors have the option to publish the peer review history of their article (what does this mean?). If published, this will include your full peer review and any attached files.

Reviewer #1: **Yes: **PAULO HENRIQUE ARAÚJO SOARES

---

## [Editor Report · Acceptance letter]

26 Dec 2024

PONE-D-24-30341R1 

PLOS ONE

Dear Dr. Nagendran, 

I'm pleased to inform you that your manuscript has been deemed suitable for publication in PLOS ONE. Congratulations! Your manuscript is now being handed over to our production team.

Kind regards, 

on behalf of

Dr. Vinícius Silva Belo 

Academic Editor

PLOS ONE